# Role of Engineered Carbon Nanoparticles (CNPs) in Promoting Growth and Metabolism of *Vigna radiata* (L.) Wilczek: Insights into the Biochemical and Physiological Responses

**DOI:** 10.3390/plants10071317

**Published:** 2021-06-28

**Authors:** Gyan Singh Shekhawat, Lovely Mahawar, Priyadarshani Rajput, Vishnu D. Rajput, Tatiana Minkina, Rupesh Kumar Singh

**Affiliations:** 1Plant Biotechnology and Molecular Biology Laboratory, Department of Botany, Jai Narain Vyas University, Jodhpur 342001, India; shrilovelymahawar@gmail.com; 2Academy of Biology and Biotechnology, Southern Federal University, 344090 Rostov-on-Don, Russia; priyadarshanirajput22@gmail.com (P.R.); rajput.vishnu@gmail.com (V.D.R.); tminkina@mail.ru (T.M.); 3Centro de Química de Vila Real, Universidade de Trás-os-Montes e Alto Douro, Quinta de Prados, 5000-801 Vila Real, Portugal; rupeshbio702@gmail.com

**Keywords:** carbon nanoparticles, nano-fertilizers, *Vigna radiata*, oxidative stress, antioxidants

## Abstract

Despite the documented significance of carbon-based nanomaterials (CNMs) in plant development, the knowledge of the impact of carbon nanoparticles (CNPs) dosage on physiological responses of crop plants is still scarce. Hence, the present study investigates the concentration-dependent impact of CNPs on the morphology and physiology of *Vigna radiata*. Crop seedlings were subjected to CNPs at varying concentrations (25 to 200 µM) in hydroponic medium for 96 h to evaluate various physiological parameters. CNPs at an intermediate concentration (100 to 150 µM) favor the growth of crops by increasing the total chlorophyll content (1.9-fold), protein content (1.14-fold) and plant biomass (fresh weight: 1.2-fold, dry weight: 1.14-fold). The highest activity of antioxidants (SOD, GOPX, APX and proline) was also recorded at these concentrations, which indicates a decline in ROS level at 100 µM. At the highest CNPs treatment (200 µM), aggregation of CNPs was observed more on the root surface and accumulated in higher concentrations in the plant tissues, which limits the absorption and translocation of nutrients to plants, and hence, at these concentrations, the oxidative damage imposed by CNPs is evaded with the rise in activity of antioxidants. These findings show the importance of CNPs as nano-fertilizers that not only improve plant growth by their slow and controlled release of nutrients, but also enhance the stress-tolerant and phytoremediation efficiency of plants in the polluted environment due to their enormous absorption potential.

## 1. Introduction

Nanotechnology is a leading field of science which involves the manipulation of material at the nanoscale (1 to 100 nm in size) to create functional materials that acquire peculiar properties over their bulk materials. At the nanoscale, carbon-based nanomaterials (CNMs), including fullerenes, nanodots, nanoparticles, nanotubes, nano-horns, nanobeads, nano-diamonds and nanofibers [1], possess novel physiochemical properties such as small surface area, increased chemical reactivity, increased ability to penetrate biological cells and typical surface morphology. These special properties vary from their bulk materials due to the differences in agglomeration shape, small size and surface structure [2], as well as due to the molecular stability of constructive CNMs and their homogeneous dispersive character in the application medium [3]. Carbon-based nanomaterials (CNMs) are explored as drug carrier vehicles and smart delivery systems in several areas of human endeavor, such as nano-pharmacology, nanomedicine, public health, etc. [4,5], for delivery of an appropriate dosage of drugs or other active substances to the specific target site within the cell [3]. Similar functions are applicable to plant systems in which CNMs are used as pesticides, growth enhancers, seed sprouts and carriers (DNA, phytohormones, fertilizers, herbicides) to the plant cells [3]. However, CNM applications in plant science, especially in sustainable crop production, have not been thoroughly explored, and hence the impacts of CNMs on plant development are less studied in comparison to the corresponding research on animals [6]. Moreover, certain studies have raised questions about the potential use of nanoparticles in plants to enhance the agricultural productivity regarding their negative impacts on living organisms and surroundings. 

The release of carbon nanoparticles into the terrestrial environment through atmospheric deposition, agricultural application, surface runoff or other pathways will accumulate CNPs in higher concentrations in soil because of their poor migration in soil [7,8]. Plants are cornerstones of all ecosystems and play an essential role in the fate and transport of CNPs in the environment via uptake (through foliar or root pathways) and bioaccumulation through the food chain [9]. The increased accretion of nanoparticles in the plant tissue affects the plant growth and physiology by inhibiting seed germination, suppressing plant elongation, decreasing plant biomass, altering gene expression and increasing the production of reactive oxygen species (ROS), such as hydrogen peroxide (H_2_O_2_ ), singlet oxygen (^1^O_2_) and hydroxyl radical (OH^−^), that induces oxidation of nucleic acids, proteins, lipids and poses a threat to the bio-membrane, which finally leads to cell death [7,8]. To overcome the negative impact of CNPs, plants have a well-developed antioxidant system which comprises of several non-enzymatic (proline, carotenoids, thiols) and enzymatic antioxidants (ascorbate peroxidase, catalase, superoxide dismutase, guaiacol peroxidase, glutathione reductase, heme-oxygenase), that can scavenge the surplus ROS. Heme-oxygenase is a novel discovered enzyme in higher plants that works as an antioxidant against different environmental stress, namely salinity, UV-B and heavy metal stress [10,11,12,13]. The enzyme catalyzes the oxidative degradation of Fe (III) protoporphyrin IX to biliverdin IX (BV), carbon monoxide (CO) and iron in the presence of a reducing equivalent (NADPH) [14,15,16]. These antioxidants show regular activity during normal conditions, but their catalytic reaction is magnified under changing environments [10]. 

Recently, Li et al. [17] reported that most nanoparticles show adverse effects on crop development at low dosage and positive impacts at high concentrations, which may differ upon changing the morphology, dosage, covering and composition of nanoparticles [17]. Toxic effects of nanoparticles, including root growth inhibition [18,19,20], formation of reactive oxygen species and rise in peroxidation of membrane lipids [21], have been well-studied in previous research. Conversely, Hong et al. [22] and Yang et al. [23] evaluated the function of TiO_2_ nanoparticles in promoting plant growth by improving their nitrogen fixation ability. Lu et al. [24] recorded the improved synthesis of nitrate reductase in *Glycine max* when subjected to TiO_2_ and SiO_2_ nanoparticles, which in turn promotes plant growth and seed germination by increasing the water uptake efficiency. Though research in this multidisciplinary field is productive, the scientific research focusing on the impact of carbon-based nanoparticles (CNPs) on plant responses is still scarce. In view of this, the present study highlights the concentration-dependent effect of engineered carbon nanoparticles (CNPs) on crop plants. 

To assess this, we studied the interaction between carbon nanoparticles and crop in liquid medium by focusing on the physiological (protein, chlorophyll, oxidative damage and antioxidants) and morphological (plant height and biomass) parameters. These parameters were selected based on the effect of uptake and accumulation of nanoparticles on these physio-morphological parameters, mainly photosynthesis, plant growth and biomass [25]. *Vigna radiata* was selected as the experimental crop to examine the effect of CNPs dosage, due to its ability to adapt in different environmental conditions. *Vigna radiata* is an economically important legume crop around the globe and among the important summer legumes, grown predominantly under semi-arid conditions of tropical and subtropical regions. The short lifecycle of the plant facilitates the study of the impact of CNPs in a relatively shorter time period [26,27]. The study is significant as it is helpful to recognize the allowable concentration of CNPs on crops so new biotechnological approaches can be developed for plant improvement.

## 2. Results 

### 2.1. Characterization of CNPs

The characterization of CNPs using UV-Vis spectroscopy is summarized in Figure 1A. The UV-Vis absorption spectrum (the red line) of CNPs solution shows an absorption peak around the 215 nm wavelength, while the image obtained from SEM predicts spherical carbon nanoparticles of varying sizes, ranging between 3 and 10 nm (Figure 1B). 

### 2.2. Effect of CNPs on Growth Parameters, Photosynthetic Pigment and Protein Content

Growth parameters, mainly plant height and biomass in the present study, were considered as primary indicators to assess the effect of nanoparticles on crop development. To infer the impact of CNPs on *Vigna radiata*, seedlings were exposed to nanoparticles for 96 h under controlled conditions in liquid medium. As shown in Figure 2B, both root and shoot length remained constant at initial concentrations, until 75 µM, but a minute increase was recorded at 100 µM, which again decreased at a higher concentration (200 µM). At 100 µM CNPs treatment, 16.65% and 5.67% increases in shoot and root lengths were observed (Figure 2B). Biomass of seedlings subjected to CNPs for 96 h increased until 100 µM, and a significant reduction was observed at elevated concentrations (150 and 200 µM). The fresh and dry weight maximums were recorded at 100 µM, which were about 1.20 and 1.14 times higher than the control (Figure 2A). Moreover, a similar pattern of results was observed for tolerance index (TI) and leaf water content (LWC), which were noted to rise with improved dosage of carbon nanoparticles up to 100 µM and then decline with a subsequent rise in dosage. The maximum values of TI and LWC were recorded at the 100 µM treatment, which were 1.26 and 1.12 times higher than the control (Figure 2C). 

The chlorophyll content in the present study increased progressively with the increase in CNPs concentration up to 100 µM, and further, a slight decrease was recorded at higher concentrations (150 and 200 µM). The maximum chlorophyll content was documented at 100 µM, which was 1.9 times higher than untreated crops (Figure 2D). The protein content in leaves of *V. radiata* after 96 h of CNPs treatment showed a noticeable increase at 25 µM, whereas at other concentrations, the protein content increase was statistically indistinguishable as compared to that of the control. No significant change was recorded in roots when exposed to varying concentrations of CNPs (Figure 2E). 

### 2.3. Effect of CNPs on Stress Parameters

Oxidative damage was determined by estimating the amount of hydrogen peroxide and malondialdehyde (MDA) production [28]. In the present work, MDA content was recorded to decline with improved dosage of CNPs up to 100 µM in both tissues. A progressive rise was documented at the 150 µM dosage, which was 1.09 (leaves) and 1.06 (roots) times higher than untreated seedlings. Further elevated concentrations of CNPs resulted in insignificant changes in MDA content (Figure 3A). The improved level of H_2_O_2_ was mostly documented in root tissue of CNP-exposed seedlings (Figure 3B). The H_2_O_2_ content initially increased (25 µM) and then remained constant until 100 µM, but a significant increase from 7.9% (100 µM) to 15.3% (150 µM) over a period of 4 days was recorded in roots. Whereas in leaves, a progressive increase in H_2_O_2_ content, 17.24% (from 25 to 50 µM), was observed, which gradually decreased thereafter with further increases in the concentration of CNPs (Figure 3B). 

### 2.4. CNPs Effect on Proline Accretion 

Accretion of proline is the common physiological alterations in plant cells subjected to changing environmental conditions. In the present study, improved amounts of proline were recorded in CNP-exposed plants compared to untreated ones, as specified in Figure 4A. There was a noticeable rise in the amount of proline with the increase in treatment of CNPs up to 100 µM. After this concentration, proline content gradually decreased with subsequent rises in dosage. The greatest accumulation of proline was recorded at 100 µM CNPs, which were 1.48 (leaves) and 6.29 (roots) times higher than in untreated seedlings (Figure 4A). 

### 2.5. Impact of CNPs on the Activity of Antioxidant Enzymes 

Superoxide dismutase (SOD) activity was found to decrease at the initial concentration of CNPs (25 µM) in both tissues. Further increases in concentration resulted in insignificant changes in SOD activity. However, no noticeable change in SOD activity was recorded until 100 µM concentration, but a progressive increase (44% more than 100 µM) was observed at 150 µM concentration in roots (Figure 4B). Improved catalysis of superoxide dismutase, with a rise in CNPs dosage, possibly signifies the augmented level of reactive oxygen species that leads to the rise in expression of the SOD gene [29].

Ascorbate peroxidase (APX) neutralizes reactive oxygen species by catalyzing the conversion of hydrogen peroxide into water by accepting electrons from ascorbate [30]. In a recent study, APX activity was recorded more in roots than foliar tissues. APX activity increased with the increase in CNPs treatment up to 100 µM, and then declined with the subsequent rise in dosage in leaves. At 100 µM, a 56.03% increase in catalysis of ascorbate peroxidase was noted in comparison to untreated seedlings in leaves (Figure 4C).

The catalytic reaction of catalase was greater in leaves than roots (Figure 4D). In leaves, an improved CAT reaction was documented with elevated concentrations of CNPs until 150 µM, which then declined with the subsequent rise in dosage. A noticeable increase in enzyme activity was noted at 100 µM, which was 1.78 times higher than in untreated seedlings in leaves. 

GOPX, the other ROS scavenging enzyme, is distinguished from APX based on substrate utilization (GOPX utilizes guaiacol while APX utilizes ascorbate as a substrate), and differences in sequences and physiological functions [30]. The catalytic reaction of GOPX was mostly observed in roots compared to leaves. The improved GOPX catalysis was noted with the rise in CNPs dosage up to 75 µM, and then declined at 100 µM. A substantial rise in GOPX activity was documented at 150 µM CNPs treatment (2.65 times higher than control), which again decreased at elevated concentrations (Figure 4E). 

Heme-oxygenase activity in the current study increased with the improved dosage of CNPs from standard to 100 µM, and further decreased in both tissues (leaves and roots). The threshold value of HO catalysis was documented at 100 µM, which was 1.29 (leaves) and 1.17 (roots) times higher than untreated seedlings (Figure 4F).

## 3. Discussion

Carbon nanoparticles (CNPs) are composed of pure carbon and show high stability, superior conductivity (both electrical and heat), high mechanical properties, environmental friendliness and low toxicity [31], and hence their usage in agriculture to increase the yield and production of crop plants is significant to feed the growing population [32]. In the present study, an attempt has been made to understand the importance of carbon nanoparticles in some of the vital physiological processes of plants by using *Vigna radiata* as a plant model. The results demonstrated that interaction of CNPs with *V. radiata* seedlings causes several morpho-physiological changes in crop plants, depending on the physiochemical properties of CNPs. Since the effect of NPs on plant tissues and their ability to penetrate them strongly depend on the physiochemical characters of NPs [3], hence the characterization of engineered CNPs is important. The UV-Vis characterization verified that CNPs possess an absorption spectrum at around 215 nm, which might originate due to π → π* activation of aromatic carbons on the carbon core, while the SEM image shows agglomeration and the spherical nature of CNPs with an average diameter of 6.5 nm (Figure 1B). The minute size of CNPs increases their penetration into the plant tissues as the penetration totally depends on the size and concentration of CNPs [32]. 

Improved CNPs penetration is the key for growth of plants [32]. In the current study, the optimum concentration of CNPs which favors plant growth is 100 µM, where all the growth parameters were found to be higher (Figure 2A,B). Further increases in CNPs treatment result in an overall reduction in plant growth. These outcomes were in accordance with the study conducted by Li et al. [17], which shows a dose–response effect of fluorescent carbon dots on the morphology (crop height and weight) of *V. radiata*. The study showed that shoot and root length increased up to a concentration of 0.4 mg/mL of fluorescent carbon dots and further declined at higher concentrations [17]. The improved growth in *V. radiata* seedlings at 100 µM CNPs is due to the fact that CNPs application increases the nitrogen and potassium content in plant organs, which is beneficial for the growth [33]. Similar enhancements in growth rate due to increases in N and K content were recorded in *N. tabaccum* and *T. aestivum* subjected to variable concentrations of CNPs, which were significantly higher in comparison to the growth obtained by the use of conventional fertilizers [6,33]. The decline in crop growth at a higher concentration (200 µM) might be due to the accumulation of carbon nanoparticles on the roots as they are directly associated with nanoparticles which restrain the uptake of minerals by the crop, and hence limit the plant growth at improved CNPs concentration (200 µM) [34].

Besides plant growth, the impact of CNPs dosage on crops is mainly reflected in the alterations in photosynthetic pigments, as chlorophyll content is regarded as one of the significant determinants of plant growth and is used as an indicator of nanoparticles’ toxicity to plants [34]. In our study, the chlorophyll content of treated *Vigna radiata* seedlings was higher at almost all concentrations, suggesting that CNPs facilitate chlorophyll biosynthesis. A similar study conducted by Wang et al. [35] demonstrated an enhancement in the photosynthetic process by carbon dots in mung bean sprouts. He illustrated that chlorophyll content was improved by 14.8% in treated crops compared to the standard. This is probably due to the fact that carbon nanoparticles enhance the photosystem activity (Rubisco activity and chlorophyll content) by accelerating the electron transfer rate [35]. In another study, a significant improvement in chlorophyll content (46.4%) and leaf protein (96%) was observed in *Vigna radiata* by an application of TiO_2_ NPs [36]. However, the minute decline in chlorophyll concentration at improved dosages (150 and 200 µM CNPs) might be due to the yellowing of leaves in CNP-exposed crops, compared to untreated crops [18]. Oxidative damage takes place in the cell’s plastid due to the probable association of ENPs with the chloroplast, that ultimately results in disruption of chlorophyll biosynthesis or causes chlorophyll reduction in leaves at higher CNPs concentrations [34]. 

Protein content initially increased in the leaves with increasing concentration of CNPs in the nutrient media and decreased at higher concentrations (Figure 2E). Elevated protein content at low CNPs concentration is recognized in the stimulation of stress proteins [10]. The comparable study with ZnO NPs conducted by Raliya and Tarafdar [37] on *Cyamopsis tetragonoloba* demonstrated that ZnO NPs enhanced the root area (73.5%), root length (66.3%), shoot length (31.5%), plant biomass (27.1%), total protein (27.1%) and chlorophyll content (276.2%).

Carbon nanoparticles treatment at a higher concentration (200 µM) in the liquid medium increases toxic effects in plants. The toxicity mechanisms of engineered nanoparticles are not known, and therefore production of ROS (H_2_O_2_, OH^−^, O_2_^−^ and O_2_^−2^) and oxidative damage were regarded as indicators to explicate the phytotoxic effect of CNPs [18]. Hydrogen peroxide is an important element that controls the defense mechanisms, acclimatory processes, metabolic processes and gene expression in plants. It is the most stable form of ROS and therefore plays a vital role as a signaling molecule in various physiological processes [18]. An increase in H_2_O_2_ content was recorded at the initial CNPs treatment (25 µM), which decreased with the rise in concentration up to 100 µM and again increased at elevated treatments (150–200 µM) (Figure 3B). Augmented generation of H_2_O_2_ leads to the surplus production of malondialdehyde [38]. The outcome of the study indicated the decline in MDA content in seedlings of *V. radiata* with the increase in CNPs concentration up to 100 µM. Comparable outcomes were noticed in maize seedlings exposed to TiO_2_ and SiO_2_ NPs, in which MDA content significantly decreased with the rise in nanoparticle concentration [34]. Contradictory results of lipid peroxidation were recorded in *Brassica juncea* treated with zinc oxide nanoparticles [18], CuO NP-treated chickpea [39] and Indian mustard seedlings [40], where MDA content progressively increased with the increase in concentration of nanoparticles (from 200 to 1000 mg/L). The reduction in oxidative damage at 100 µM concentration might be due to the small size of CNPs (3–10 nm), which have a higher penetration ability and can easily penetrate in plant organs and increase the nutrient absorption and nutrient flow. The increased intracellular nutrient content increases the tolerance as well as adaptation ability of plants in different environmental conditions, and thereby decreases the oxidative damage (MDA and H_2_O_2_ content) [3]. Moreover, the cell wall acts as a preliminary barrier for the penetration of nanoparticles into the plant cell. Parallel outcomes were reported in maize seedlings, where TiO_2_ NPs were only recorded on the surface of leaves and no cellular penetration of nanoparticles was noticed [34]. A progressive increase in MDA and H_2_O_2_ content after 100 µM CNPs treatment indicates that CNPs mediated oxidative stress induction at higher concentrations. Similar results were verified in *Oryza sativa* exposed to CuO nanoparticles [41]. The direct association of plant cells with copper oxide nanoparticles or surplus copper ions results in the improved production of reactive oxygen species and malondialdehyde in *Brassica juncea* [40]. 

The contradictory results of lipid peroxidation and proline with the increase in CNPs dosage in the study indicate a linear relation between the formation of reactive oxygen species (hydrogen peroxide) and its mitigation via proline. Besides operating as an osmolyte, proline also work as a scavenger that quenches the metal ions and reactive oxygen species intermediates (hydroxyl radicals and singlet oxygen) and confers a shield towards stress-stimulated cell destruction [18,40]. Increased proline amount with elevated CNPs concentration specifies the change in membrane permeability, resulting in water stress that finally leads to higher proline content. Proline accumulation in the present work was comparable with the preceding reports on diverse crops with different environmental conditions [10,29,42,43,44,45]. 

Disproportion in reactive oxygen species metabolism is the principal cause of plant cell damage. CAT, GOPX, APX, SOD and HO are the chief antioxidant enzymes that guard biological cells by eradicating hydrogen peroxide and other ROS [46]. During reactive oxygen species detoxification, the preliminary reaction was initiated by superoxide dismutase, which provides defense against the toxic effects of ROS by deactivating superoxide radicals into H_2_O_2_ and O_2_ [47]. H_2_O_2_ generated by the superoxide dismutase activity was subsequently reduced to H_2_O by utilizing other enzymes such as GOPX, APX and CAT [10]. In our study, the activity of all antioxidants, including SOD, CAT, POD, APX and HO, increased under CNPs treatment, which shows the clear response of *V. radiata* seedlings towards carbon nanoparticles, although the decrease in enzyme activity at 200 µM might be due to the phytotoxic effect of CNPs on protein formation and other physiological parameters at high concentrations that inhibit enzyme proteins [18]. Similar trends in antioxidant activity were recorded in *Brassica juncea* and *Brassica nigra* treated with ZnO [18] and Ag nanoparticles [48]. Moreover, we observed dissimilar antioxidant activity in our study, such as increased activity of peroxidases (GOPX and APX) in roots and CAT in leaves after exposure of CNPs treatment for a period of 96 h. The difference in antioxidant activity in crop tissues demonstrates that these antioxidants were working in parallel to exterminate hydrogen peroxide. The insignificant activity of catalase in roots was indemnified by the improved catalysis of peroxidases. Peroxidase has stronger reactivity for hydrogen peroxide (in µM range) than catalase (mM range) and plays a significant function in neutralizing ROS under a changing environment [49]. Moreover, catalase is reactive towards superoxide radicals, and hence their growing amount at higher CNPs treatment may result in the inactivation of enzymes [50]. The reduction in enzymes’ activity might also be associated with damage by peroxisomal proteases or probably due to inactivation of antioxidants [51]. 

Exposure of CNPs on plants and their interaction causes several physiological changes in the plant species, which depend on the properties of CNPs such as shape, size, dosage, reactivity, surface covering, type and chemical composition. The efficiency of particular types of CNPs varies with the plant, and the effect (either beneficial or adverse) of the CNPs–plant interaction, is always concentration-dependent [3]. In our study, intermediate concentrations (100 µM) of CNPs enhanced the growth of *V. radiata* by increasing plant biomass, chlorophyll content, augmentation of antioxidants and lowering the oxidative damage. This might be due to the higher penetration of CNPs, which is inversely proportional to size and is the key factor behind improved growth. The carbon nanoparticles are taken-up by plants from the liquid medium via roots and distributed in the aerial parts through capillary action [52]. The absorption and distribution of CNPs into the plant system results in remarkable changes in metabolic functions, including increasing water uptake (by inducing expression of gene-encoding water channel proteins or by cell wall pores), and enhancing nutrient (Fe, Mg, Ca and K) absorption, leading to an increase in plant biomass and improved physiological activity [52] (Figure 5). Treatment with ENPs at higher concentrations (above 100 µM) might increase the amounts of nanoparticles in crop tissues, which seem to cross the bio-membrane and assemble to form clumps with their own particles or with biomaterials present inside the cell [53]. This might create a disturbance in the cellular compartment and increase the hydroxyl radical or other forms of ROS inside the cell, which finally results in an increase in the activity of antioxidants. The activated antioxidant mechanisms control and establish the redox balance, but at elevated concentrations (200 µM), biochemical effects do occur. Elevated concentration of all the antioxidant molecules is the possible reason for the high tolerance level exhibited by *V. radiata* to CNPs. Moreover, at elevated treatment (200 µM), higher CNPs concentrations might accumulate in the plant tissues and block the passage for nutrients to flow further, and thus hamper the plant growth [52]. 

## 4. Materials and Methods

### 4.1. Carbon Nanoparticles (CNPs): Synthesis and Characterization 

Carbon nanoparticles (CNPs) were synthesized by the microwave-assisted method [54]. Sucrose solution (in water) was mixed with ortho-phosphoric acid (88% *v/v*) by heating it in a microwave for 5 to 15 min. When the color of the solution changed from yellow to brownish black, distilled water was added to it after cooling. To obtain the carbon nanoparticles (CNPs), the solution was centrifuged for 10 min at 4000× *g*. The procured CNPs precipitates were washed 5 to 6 times with deionized water, followed by centrifugation to remove the remaining traces of acid. The purified CNPs were filtered on Whatman filter paper No. 1. The filter papers containing CNPs were dehydrated overnight in an oven. After drying, nanoparticles were scraped out and stored in an air-tight container (Figure 6). The nanoparticles were dispersed in distilled water (pH ≥ 7) for further use [54]. The CNPs were characterized using a UV-Visible spectrophotometer to determine the optical properties of a solution. For examination of nanoparticles, a 1 mM solution of carbon nanoparticles was prepared and analyzed for the wavelength from 200 to 650 nm. Moreover, the morphology, dimension and structure of CNPs were determined using a scanning electron microscope.

### 4.2. Plant Cultivation 

*Vigna radiata* var. *PDM 54* seeds were obtained from NBPGR, Jodhpur, India, and stored in an air-tight container. Before use, the seeds were disinfected with 0.1% HgCl_2_ for thirty seconds to one minute to avoid contamination, and cleansed three to four times with deionized water to make sure they were free from any traces of mercuric chloride. These seeds were then germinated in a sterile environment in a seed germinator at 25 °C in the dark. After germination, identical *V. radiata* seedlings were transferred to pots (12 × 12 cm^2^) containing Hoagland nutrient solution (pH 6.8 to 6.9) and placed at 25 ± 2 °C, 50% relative humidity, to set-up the liquid culture. The liquid medium was aerated twice daily to avoid inadequate supply of oxygen and prevent precipitation of salts. The nutrient medium was replaced every alternate day to circumvent nutrient deficiencies.

### 4.3. Carbon Nanoparticles Treatment

After seven days, acclimatized *V. radiata* seedlings in liquid medium were exposed to CNPs at different dosages, from 25 to 200 µM (25, 50, 75, 100, 150 and 200 µM). A nutrient medium without CNPs was regarded as the control and utilized to evaluate the effect of CNPs on crops. After 96 h, seedlings of uniform size were harvested for the analysis of several morphological and physiological characteristics [10].

### 4.4. Growth Analysis 

For growth parameter studies, freshly harvested crop seedlings were cleansed with deionized water and roots were separated from aerial tissue. Growth of the crops was analyzed by measuring plant height (root and shoot length), biomass (fresh and dry weight), leaf water content and tolerance index. Plant height of both treated and control seedlings was measured in centimeters. To determine the dry weight, fresh seedlings were desiccated at 65 °C overnight in an oven, and the weight of dehydrated seedlings was measured. The water content of the leaves was computed by the equation: (fresh weight – dry weight)/fresh weight × 100. Tolerance index was calculated according to the method of Wilkins [55] and expressed in percentage. 

### 4.5. Estimation of Photosynthetic Pigment and Protein Content

Chlorophyll contents (chlorophyll a, b and total chlorophyll) were estimated by Arnon’s [56] method. Fresh leaves of the seedlings were pulverized in chilled acetone (80% *v/v*). Grounded samples were centrifuged in cold conditions for 15 min at 10,000× *g*. The amount of chlorophyll (mg g^−1^ fresh weight of leaves) was computed from the optical density of the supernatant recorded at 663 and 645 nm [56]. 

Protein content was determined via Lowry et al.’s [57] method. Freshly harvested tissue was ground in sodium phosphate buffer (50 mM, pH 7.0). The procured supernatant after centrifugation (at 4 °C for 10 min at 10,000× *g*) was reacted with an assay mixture (Na_2_CO_3_ (2% *w/v*) in NaOH (0.1 N) + CuSO_4_·5H_2_O (0.5% *w/v*) + KNaC_4_H_4_O_6_·4H_2_O (1%)) at room temperature for 10 min. After incubation, Folin–Ciocalteu reagent was incorporated in the above reaction mixture. The optical density of the blue color complex formed after 30 min of incubation was noted at 660 nm. The protein content (mg g^−1^ fresh weight of tissue) was computed from the linear curve prepared by utilizing bovine serum albumin (BSA) as a control.

### 4.6. Determination of Stress Parameters (Lipid Peroxidation and H_2_O_2_ Content)

Peroxidation of lipid was estimated by De Vos et al.’s [58] protocol by measuring the amount of malondialdehyde (MDA). Plant tissue was crushed in 2-thiobarbituric acid (0.25% *w/v*) made in trichloroacetic acid (10% *v/v*). The sample was reacted for half an hour in a water bath and further brought to room temperature to cease the reaction. The reacted sample was centrifuged for 10 min at 3000× *g*. The amount of peroxidation of lipid (nm g^−1^ fresh weight of tissue) was calculated from the specific absorbance (λ_532_ – λ_600_) by utilizing 155 mM^−1^ cm^−1^ as the molar absorption coefficient.

H_2_O_2_ content in crop seedlings was studied spectrophotometrically by Alexivea et al.’s [59] method. Fresh plant tissue, homogenized in trichloroacetic acid (0.1% *w/v*) in cold conditions, was centrifuged for 15 min at 10,000× *g*. Absorbance of the supernatant reacted with KPO_4_ buffer (10 mM, pH 7) and KI (1 M) for an hour in the dark was documented at 390 nm. H_2_O_2_ level was estimated by utilizing 0.28 µM^−1^ cm^−1^ as the proportionality coefficient. 

### 4.7. Antioxidative Response Evaluation

Proline content was estimated according to Bates et al.’s [60] protocol. Fresh tissue homogenized in sulfosalicylic acid (3% *w/v*) was centrifuged for 20 min at 3000 × *g*. The procured supernatant was incubated with equal volumes of ninhydrin solution and glacial acetic acid at 100 °C for an hour and immediately transferred on ice. Absorbance of the colored organic layer obtained by the addition of toluene to the cooled reaction mixture was documented at 520 nm. The amount of proline (µg g^−1^ fresh weight of tissue) was computed from a linear graph made by L-proline. 

### 4.8. Enzymatic Assay 

Fresh plant tissue homogenized in NaPO_4_ buffer (50 mM, pH 7) was centrifuged at 4 °C for 20 min at 5000× *g*. The supernatant obtained was utilized for the enzymatic assay.

Ascorbate peroxidase (APX) activity was assayed according to Chen and Asada’s [61] protocol. The rate of oxidation of ascorbic acid was documented by the decline in optical density of the assay mixture (NaPO_4_ buffer 50 mM, pH 7, having ascorbic acid (0.6 mM) + hydrogen peroxide (10% *v/v*) + enzyme extract) at 290 nm by utilizing 2.8 mM^−1^ cm^−1^ as the proportionality constant [61].

The catalytic reaction of catalase (CAT) was analyzed via Aebi’s [62] procedure. The degradation rate of hydrogen peroxide was noted by the decline in optical density of the reaction compound (NaPO_4_ buffer 50 mM, pH 7 + hydrogen peroxide (9 mM) + enzyme extract) at 240 nm by inserting 0.039 mM^−1^ cm^−1^ as the proportionality constant [62].

The guaiacol peroxidase (GOPX) catalytic reaction was evaluated by documenting the rise in optical density of the assay compound (NaPO_4_ buffer 50 mM + hydrogen peroxide (3.7 mM) + guaiacol (20 mM) + enzyme extract) at 436 nm (molar absorption coefficient 26.6 mM^−1^ cm^−1^) [63].

The catalytic reaction of superoxide dismutase (SOD) was estimated by Beauchamp and Fridovich’s [64] method. The activity was assayed by recording the optical density of the assay compound (NaPO_4_ buffer 50 mM, pH 7 + NBT (75 µM) + riboflavin (2 mM) + methionine (13 mM) + EDTA (0.1 mM) + enzyme extract) at 560 nm after half an hour of incubation under bright light, which will quantify the capability of an enzyme to hinder the photochemical reduction of NBT (nitroblue tetrazolium) [64]. 

Heme-oxygenase (HO) catalysis was assayed via Balestrasse et al.’s [65] protocol. Freshly harvested tissue homogenized in chilled buffer (KPO_4_ buffer, 50 mM, pH 7.4 + EDTA (200 µM) + PMSF (1000 µM) + sucrose (250 mM)) was centrifuged for 20 min at 20,000× *g* in a cold environment. The supernatant (HO extract) was reacted with NADPH (0.06 µM) and hemin (0.2 µM) for an hour at 37 °C, and the absorbance of the obtained product (biliverdin) (proportionality constant 6.25 µM^−1^ cm^−1^) was recorded at 650 nm.

### 4.9. Examination of Data

Results were statistically analyzed by SPSS 16. Data were regarded as average (± standard deviation) of independent replicas (*n* = 3). The variations between standard and concentrations were examined by utilizing one-way ANOVA (analysis of variance) at the 0.05% significance level via Duncan’s multiple range test (DMRT) [10]. 

## 5. Conclusion

From the present work, it is concluded that CNPs at intermediate concentrations (100–150 µM) favor the growth of *V. radiata* seedlings. At these concentrations, the highest activity of antioxidants (SOD, GOPX, APX and proline) were recorded, which resulted in the decline of the ROS level, and hence, the total biomass increased. At the highest treatment (200 µM), the aggregation of CNPs was observed more on the root surface, as the root comes in direct contact and gets accumulated in higher concentrations in the plant tissues, which blocks the passage for nutrients and thus inhibits the uptake and translocation of nutrients to plants. Hence, the oxidative damage enforced by CNPs circumvents with the improved activity of antioxidants. The increases in biomass, total chlorophyll and protein content in CNP-treated *V. radiata* seedlings in the study are valuable from an agricultural perspective. CNPs promote nutrient absorption and accumulation amount, thus increasing the efficiency of fertilizer, which finally improves plant quality. Moreover, CNPs can greatly contribute to pollutant removal and soil remediation as they possess an enormous absorption potential due to their high surface area. Hence, CNPs could be a preferable choice as nano-fertilizers, compared to conventional fertilizers or manure, that not only improve plant growth by their slow and controlled release of nutrients, but also enhance the stress-tolerant and phytoremediation efficiency of the plant in the polluted environment. 

## Figures and Tables

**Figure 1 plants-10-01317-f001:**
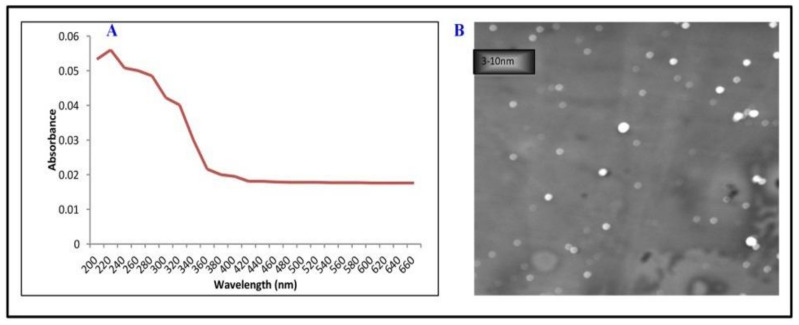
Physio-chemical characterization of synthesized carbon nanoparticles (CNPs) (**A**) UV-Vis Spectroscopy, (**B**) Scanning electron microscope (SEM) of CNPs with an average diameter of 6.5 nm.

**Figure 2 plants-10-01317-f002:**
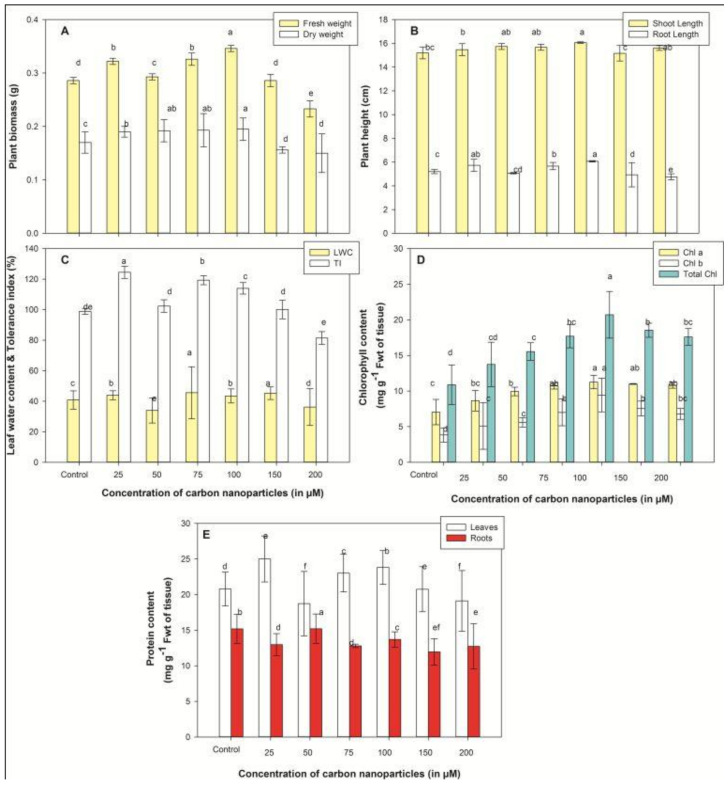
Plant biomass (**A**), plant height (**B**), leaf water content and tolerance index (**C**), chlorophyll concentration (**D**) and protein content (**E**) in seedlings of *V. radiata* treated with various concentrations of CNPs, ranging from 25 to 200 µM. Values are mean ± standard deviation (*n* = 3) and are statistically significant according to the DMRT test (*p* < 0.05). Data points marked with the same letters show insignificant differences (*p* < 0.05) within treatments.

**Figure 3 plants-10-01317-f003:**
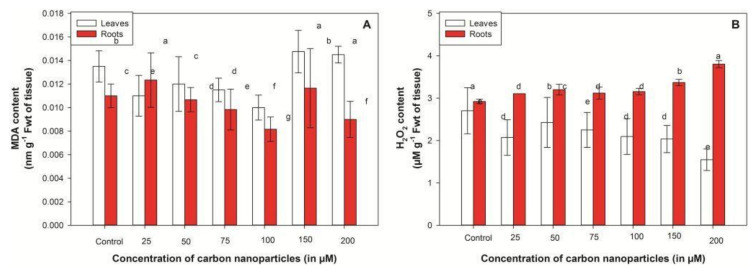
Effect of CNPs on MDA (**A**) and ROS formation (H_2_O_2_ production) (**B**) in *Vigna radiata* seedlings. Seedlings were treated with 25, 50, 75, 100, 125, 150 and 200 µM CNPs for a period of 96 h. Values are mean ± standard deviation (*n* = 3) and are statistically significant according to the DMRT test (*p* < 0.05). Data points marked with different letters show significant differences (*p* < 0.05) within treatments.

**Figure 4 plants-10-01317-f004:**
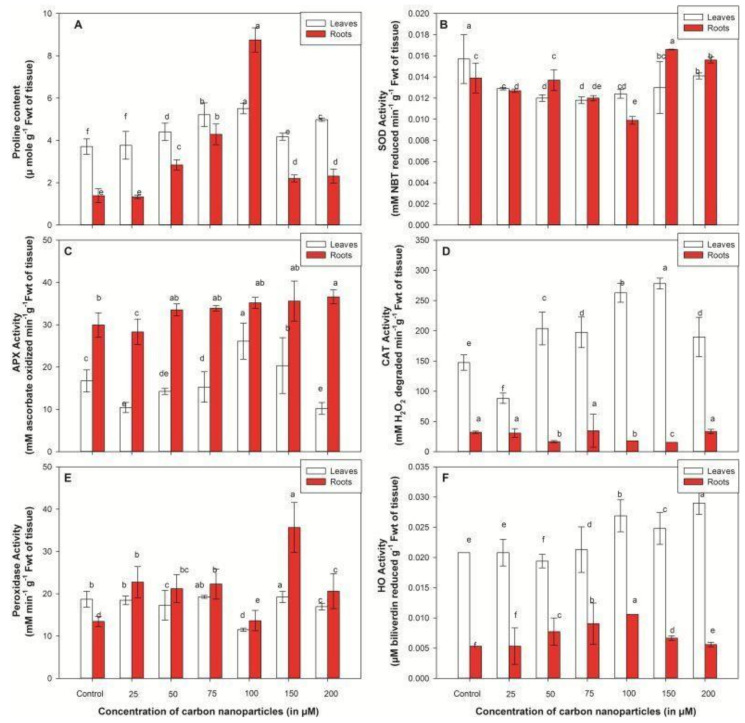
Effect of carbon nanoparticles on proline content (**A**), SOD activity (**B**), APX activity (**C**), CAT activity (**D**), peroxidase activity (**E**) and HO activity (**F**) in *Vigna radiata* seedlings at various treatments, ranging from 25 to 200 µM. Values are mean ± SD of three replicates (*n* = 3) and are statistically significant according to DMRT test (*p* < 0.05). Data points marked with the same letters show insignificant differences (*p* < 0.05) within treatments.

**Figure 5 plants-10-01317-f005:**
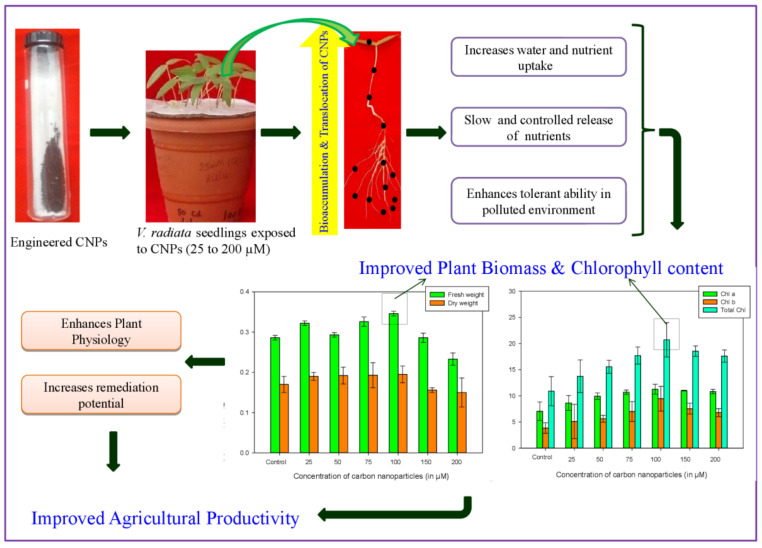
Schematic representation of the role of carbon nanoparticles (CNPs) in improving agricultural productivity of crop plants.

**Figure 6 plants-10-01317-f006:**
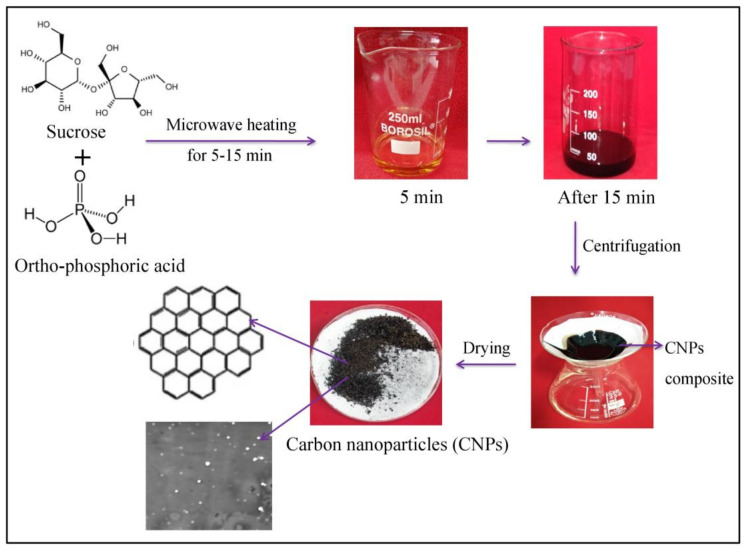
Representation of the mechanism of synthesis of carbon nanoparticles (CNPs) by heating sucrose solution and ortho-phosphoric acid.

## Data Availability

Not applicable.

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
