# Peer review of "Role of Engineered Carbon Nanoparticles (CNPs) in Promoting Growth and Metabolism of Vigna radiata (L.) Wilczek: Insights into the Biochemical and Physiological Responses"

_plants, 2021, doi:10.3390/plants10071317_

Round 1

Reviewer 1 Report

It is an average interesting work with potential to be published, after some improvement.

The English is acceptable. Some minor in corrections can be done (line 34 - carbon, line 139 – Folin, line 84 and 236 – space).

Introduction can be improve with respect to stress response mechanisms related to nanoparticles. There is a lack of the scientific explanation on the relationship of nanoparticles with oxidative stress. Some references of the reactive oxygen species detoxification should be made to a better understanding of the role of nanoparticles in the oxidative stress and the induced biochemical response (as appears in the document title).

Methods are adequately described. It should preferably be used “protein content” instead of “total protein”. Figure 1 must be improved. Some explanation should be done justifying the analytical parameters chosen for biochemical responses.

In line 158, it should preferably be used "antioxidative response evaluation” instead of “Estimation of antioxidants” because refers to enzymes and compounds involved on a global response to oxidative stress but no other that have antioxidant properties.

Results are clearly presented and support the main conclusions.

The work gives some contribution to the knowledge of the toxicity mechanisms of engineered nanoparticles but introduction and discussion can be improved with respect the plant metabolic response.

Therefore I recommended reconsider after major revision.

Author Response

Thank you 

Reviewer 2 Report

The manuscript entitled “Role of Engineered Carbon Nanoparticles (CNPs) in Promoting Growth and Metabolism of Vigna radiata (L.) Wilczek: Insights into the Biochemical and Physiological responses” depicted the effects of different CNPs concentrations on growth and antioxidant responses of Vigna radiata. The work is interesting. My suggestion is minor revision.

  1. Vigna radiata should be italic in key words.
  2. Authors should explain the reason of why choosing this variety of Vigna radiata as test materials.
  3. Line 166. Lack of the title number and title should be italic.
  4. Characterization of CNPs should be placed in Materials and methods section, not results.
  5. Line 251: P should be italic. This problem across whole manuscript.
  6. Conclusion is too long and there should be only one paragraph.

Author Response

Thank you 

Reviewer 3 Report

The research results show that 100-150 uM carbon nanoparticles (CNP) can promote the growth and the activity of antioxidant enzymes in Vigna radiata var.. The conclusion of the article is complete, but there are many contents that need to be revised. The explanation is as follows.

1. In the introduction, it is necessary to increase the correlation between CNP, antioxidant enzymes and HO activity.

2. The figures of the experimental are too rough to be able to identify the distinctive symbols. It is recommended that the figures would be redrawn.

3. The significant level symbols in the figure is so not clearly marked that cannot be identified. For example: Fig 3D, when processing 50 uM CNP, the significant level symbols of Chla, Chlb and total Chl are all marked with "e". Moreover, the significant level of the symbol may be incorrect, such as: Fig 3E, considering the mean and standard deviation, there is almost no significant difference in all treatment values, but the sign of the significant difference is marked, please reconfirm it.

4. The legend of figure 5 must be revised.

5. Please follow the result description in the order of the figure. For example, the order of Figure 5 is A. Proline content B SOD activity. C. APX activity D. CAT activity E. Peroxidase activity F. HO activity.

6. For the description of the experimental background in the results, please move to the introduction or discussion. For example

   Line 201-202,

   Line 229-230

   Line 240-242 move to discussion.

Author Response

Thank you 

Round 2

Reviewer 1 Report

All suggested changes for improvement have been made. I believe the authors followed the recommendations carefully, fulfilling the necessary requirements to improve the document. Therefore the document now submitted can be accepted for publication